# Ambroxol Treatment Suppresses the Proliferation of *Chlamydia pneumoniae* in Murine Lungs

**DOI:** 10.3390/microorganisms9040880

**Published:** 2021-04-20

**Authors:** Dávid Kókai, Dóra Paróczai, Dezső Peter Virok, Valéria Endrész, Renáta Gáspár, Tamás Csont, Renáta Bozó, Katalin Burián

**Affiliations:** 1Department of Medical Microbiology and Immunobiology, University of Szeged, Dóm Square 10, 6720 Szeged, Hungary; paroczai.dora@med.u-szeged.hu (D.P.); virok.dezso.peter@med.u-szeged.hu (D.P.V.); endresz.valeria@med.u-szeged.hu (V.E.); burian.katalin@med.u-szeged.hu (K.B.); 2Metabolic Diseases and Cell Signaling (MEDICS) Research Group, Interdisciplinary Center of Excellence, Department of Biochemistry, University of Szeged, Dóm Square 9, 6720 Szeged, Hungary; gaspar.renata@med.u-szeged.hu (R.G.); csont.tamas@med.u-szeged.hu (T.C.); 3Department of Dermatology and Allergology, University of Szeged, Dóm Square 10, 6720 Szeged, Hungary; bozo.renata@med.u-szeged.hu

**Keywords:** *Chlamydia pneumoniae*, ambroxol, anti-chlamydial, mouse model, extracellular signal-regulated kinase pathway, surfactant proteins

## Abstract

Ambroxol (Ax) is used as a mucolytics in the treatment of respiratory tract infections. Ax, at a general dose for humans, does not alter *Chlamydia pneumoniae* growth in mice. Therefore, we aimed to investigate the potential anti-chlamydial effect of Ax at a concentration four timed higher than that used in human medicine. Mice were infected with *C. pneumoniae* and 5-mg/kg Ax was administered orally. The number of recoverable *C. pneumoniae* inclusion-forming units (IFUs) in Ax-treated mice was significantly lower than that in untreated mice. mRNA expression levels of several cytokines, including interleukin 12 (IL-12), IL-23, IL-17F, interferon gamma (IFN-γ), and surfactant protein (SP)-A, increased in infected mice treated with Ax. The IFN-γ protein expression levels were also significantly higher in infected and Ax-treated mice. Furthermore, the in vitro results suggested that the ERK 1/2 activity was decreased, which is essential for the *C. pneumoniae* replication. SP-A and SP-D treatments significantly decreased the number of viable *C. pneumoniae* IFUs and significantly increased the attachment of *C. pneumoniae* to macrophage cells. Based on our results, a dose of 5 mg/kg of Ax exhibited an anti-chlamydial effect in mice, probably an immunomodulating effect, and may be used as supporting drug in respiratory infections caused by *C. pneumoniae*.

## 1. Introduction

Ambroxol (Ax; 2-amino-3,5-dibromo-N-(trans-4-hydroxycyclohexyl) benzylamine) is widely used in the treatment of respiratory infections, chronic bronchitis, and neonatal respiratory distress syndrome due to its mucus viscosity-altering effect and has been shown to be a relatively safe drug [1,2]. Furthermore, Ax shows beneficial effects on the course of Parkinson’s disease by modifying glucocerebrosidase chaperon activity [3]. Ax exhibits proinflammatory properties by elevating interleukin-10 (IL-10), IL-12, and interferon gamma (IFN-γ) expression; due to its aromatic moiety, Ax exhibits oxidant scavenger functions [4,5]. Moreover, it has been reported that Ax elevates surfactant protein (SP) production and can reverse the lipopolysaccharide-stimulated induction of the extracellular signal-regulated kinase (ERK) 1/2 pathway [6,7]. In addition, Ax is used for the symptomatic treatment of sore throat during viral infections, as it shows local anesthetic effects [8,9].

*Chlamydia pneumoniae*, belonging to the *Chlamydiaceae* family, is a Gram-negative obligate intracellular bacterium with a specific biphasic lifestyle. It has been shown that two species of this family that are major human pathogens (*Chlamydia trachomatis* and *C. pneumoniae*), and they are partially dependent on ERK pathway activity during their growth cycle [10,11]. As demonstrated previously by Oberley et al., SPcan enhance the engulfment of Chlamydiae by human THP-1 cells [12]. Moreover, it has been reported that *C. pneumoniae* and *C. trachomatis* interfere with host cell apoptosis in a time-dependent manner via pro- and antiapoptotic processes [10,13]. *C. pneumoniae* is a common cause of acute respiratory infections, such as pharyngitis, bronchitis, and sinusitis, and is responsible for approximately 10% of community-acquired pneumonia cases, in addition to being associated with nonrespiratory diseases, including atherosclerosis [14,15,16]. In our previous study, we demonstrated that N-acetyl cysteine, a commonly used mucolytic, increases the proliferation of *C. pneumoniae* in vitro and in vivo, and Ax treatment induces the expression of the anti-chlamydial enzyme indoleamine 2,3-dioxygenase 2 (IDO-2) at the transcriptional level in vitro. However, in vivo experiments in mice indicated that commonly used therapeutic concentrations of Ax do not affect *C. pneumoniae* proliferation [17]. Yang et al. reported that an elevated dose of 10-mg/kg Ax (eight-fold greater than the therapeutic concentration for humans) significantly decreases the mortality of mice administered with a lethal dose of the H3N2 virus [9]. Therefore, we hypothesized that a dose of Ax four-fold higher than normal may alter the *C. pneumoniae* proliferation in mouse lungs.

The aim of this study was to verify this hypothesis and to determine the underlying immunological mechanisms induced by Ax that contribute to the elimination of *C. pneumoniae* in mice.

## 2. Materials and Methods

### 2.1. Cultivation of C. pneumoniae

Human epithelial type 2 (HEp-2) cells obtained from the American Type Culture Collection (ATCC, Manassas, VA, USA) were used to cultivate *C. pneumoniae* CWL029 (ATCC), as described previously [18,19]. After partial purification and concentration, aliquots of elementary bodies (EBs) were added to sucrose-phosphate-glutamic acid buffer (SPG) and stored at −80 °C until use [20]. An indirect immunofluorescence assay was performed to determine the titer of infectious *C. pneumoniae* EBs. Ten-fold serial dilutions of the purified EBs were inoculated into McCoy cells (ECACC, London, UK), and after 48 h of incubation, the infected cells were fixed with acetone at −20 °C and stained with monoclonal anti-*Chlamydia* lipopolysaccharide antibodies (AbD Serotec, Oxford, UK) and fluorescein isothiocyanate (FITC)-labeled anti-mouse IgG (Sigma-Aldrich, St. Louis, MO, USA). Cells were analyzed using an Olympus UV microscope; the titer of *C. pneumoniae* was expressed as inclusion-forming units (IFU)/mL.

### 2.2. Infection of Mice and Preparation of Lung Tissues

Female pathogen-free BALB/c mice (6 weeks old) were purchased from Charles River Laboratories (Veszprém, Hungary). Animals were provided with food and water ad libitum and maintained under standard husbandry conditions at the animal facility of the Department of Medical Microbiology and Immunobiology at the University of Szeged. The mice were divided into four groups: untreated/uninfected, untreated/infected, treated/uninfected, and treated/infected (10 mice/group). Prior to infection, the mice were mildly sedated via an intraperitoneal injection of 200 μL of sodium pentobarbital (7.5 mg/mL), followed by the intranasal administration of 2 × 10^5^ IFU of *C. pneumoniae* in 20-μL SPG buffer.

From the first day post-infection (pi), the mice were administered daily with 5-mg/kg Ax (Sigma-Aldrich, Darmstadt, Germany) dissolved in water (2 g/L). Control mice received the same amount of water via oral administration using a pipette to mimic the stress of the treatment process. Behavior and weight were monitored daily. The mice were anesthetized and euthanized 7 days pi. The sera were harvested via cardiac puncture. The lungs were removed and homogenized using acid-purified sea sand (Fluka Chemie AG, Buchs, Switzerland). Half of the homogenized tissue sample was processed for RNA extraction and quantitative reverse-transcription polymerase chain reaction (RT-qPCR), whereas the other half was suspended in 1 mL of SPG for the detection of viable *Chlamydia* and to measure the cytokine levels.

This study was approved by the National Scientific Ethical Committee on Animal Experimentation of Hungary (August, 10 August 2016; III./3072/2016) and Animal Welfare Committee of the University of Szeged. This study conformed to the Directive 2010/63/EU of the European Parliament.

### 2.3. Culturing of C. pneumoniae from Murine Lungs

One half of the homogenized lung sample derived from individual mice was centrifuged (10 min, 400× *g*) (Sorvall^®^ RT6000B, DuPont, Wilmington, DE, USA), after which, the serial dilutions of the supernatants were inoculated into McCoy cell monolayers and centrifuged (1 h, 800× *g*). After 48 h of culturing, the cells were fixed and visualized, and the inclusions were enumerated as described previously [21].

### 2.4. Total RNA Extraction and cDNA Synthesis

The remaining half of the homogenized lung sample was processed using the TRI reagent (Sigma-Aldrich) according to the manufacturer’s protocol for performing total RNA extraction. RNA concentrations were measured at 260 nm using a NanoDrop spectrophotometer (Thermo Fisher Scientific, Waltham, MA, USA). The purity of the RNA samples was determined based on the ratio of RNA absorbance at 260 and 280 nm, which was > 2 for all samples. Subsequently, 1 μg of total RNA was reverse-transcribed using Maxima Reverse Transcriptase according to the manufacturer’s instructions using random hexamer primers (Thermo Fisher Scientific).

### 2.5. qPCR Amplification of Ido1, Ido2, Il12, Il23, Il17a, Il17f, Ifng, Sftpa, Sftpb, Sftpc, and Sftpd

qPCR was performed using a Bio-Rad CFX96 real-time system with the SsoFast™ EvaGreen^®^ qPCR Supermix (Bio-Rad, Hercules, CA, USA) and the following murine-specific primer pairs: *Ido1*: 5′-GCTTCTTCCTCGTCTCTCTATTG-3′ and 5′-TCTCCAGACTGGTAGCTATGT-3′, *Ido2*: 5′-CCTGGACTGCAGATTCCTAAAG-3′ and 5′-CCAAGTTCCTGGATACCTCAAC-3′, *Actb* encoding beta-actin: 5′-TGGAATCCTGTGGCATCCATGAAAC-3′ and 5′-TAAAACGCAGCTCAGTAACAGTCCG-3′, Ifng: 5′-CAAGTGGCATAGATGTGGAAGA-3′ and 5′-GCTGTTGCTGAAGAAGGTAGTA-3′, *Il12* p40 homodimer: 5′-ACATCAAGAGCAGTAGCAGTTC-3′ and 5′-AGTTGGGCAGGTGACATCC-3′, *Il23* p19: 5′-CCTGCTTGACTCTGACATCTT-3′ and 5′-TGGGCATCTGTTGGGTCTC-3′, *Il17a:* 5′-AAGGCAGCAGCGATCATCC-3′ and GGAACGGTTGAGGTAGTCTGAG-3′, *Il17f*: 5′-AGCAAGAAATCCTGGTCCTTCGGA-3′ and 5′-CTTGACACAGGTGCAGCCAACTTT-3′, *Sftpa:* 5′-GGTGTCTAAGAAGCCAGAGAAC-3′ and 5′-CAAGATCCAGATCCAAGGAAGAG-3′, *Sftpb:* 5′-CCACCTCCTCACAAAGATGAC-3′ and 5′-TTGGGGTTAATCTGGCTCTGG-3′, *Sftpc:* 5′-ATGGACATGAGTAGCAAAGAGGT-3′ and 5′-CACGATGAGAAGGCGTTTGAG-3′, and *Sftpd:* 5′-TCTGAGCCACTGGAGGTAAA-3′ and 5′-CAACATACAGGTCTGAGCCATAG-3’. All primers were designed using the PrimerQuest Tool software and synthesized by Integrated DNA Technologies Inc. (Coralville, IA, USA). A melting curve analysis was performed to verify the amplification specificity. Threshold cycles (Ct) were determined for *Ido1*, *Ido2*, *Il12*, *Il23*, *Il17a*, *Il17f*, *Ifng*, *Sftpa*, *Sftpb*, *Sftpc*, *Sftpd*, and *Actb*, and the relative gene expression was calculated via the 2-(ΔΔCt) method. One-way analysis of variance with repeated measures (ANOVA RM) and planned comparisons were used to compare the statistical differences in the log2(ΔΔCt) values between the infected and control samples, as described previously, with a level of significance of *p* < 0.05 [22].

### 2.6. Enzyme-Linked Immunosorbent Assay (ELISA)

Standard sandwich mouse IL-6 and mouse IFN-γ ELISA kits (BD OptEIA TM; BD biosciences, Franklin Lakes, NJ, USA) were used to determine the IL-6 and IFN-γ concentrations in the lung supernatants. Prior to use, the stored lung samples were thawed immediately, and the assay was performed according to the manufacturer’s instructions at 10× dilution for IL-6 and at 50× dilution for IFN-γ. The dynamic range of the kits was between 10 and 1000 pg/mL. Plates were analyzed using the Biochrom Anthos 2010 microplate reader (Biochrom, Cambridge, UK). Samples were assayed in duplicate.

### 2.7. Western Blotting Analysis of ERK

To investigate the levels of phosphorylation in RAF proto-oncogene serine/threonine-protein kinase (c-Raf), as well as dual specificity mitogen-activated protein kinase (MEK)1/2, ERK1/2, P90RSK1/2, and mitogen- and stress-activated protein kinase 1 (MSK-1) proteins, McCoy cells were grown in 6-well plates and treated with Ax (0.05 mg/mL) and/or infected with *C. pneumoniae* simultaneously at 0.01 multiplicity of infection (MOI) or left untreated and/or uninfected. After a 12-h incubation period, the cells were washed twice with ice-cold phosphate-buffered saline, scraped, and collected in a homogenization buffer (1× radioimmunoprecipitation buffer supplemented with phosphatase inhibitors and a protease inhibitor cocktail) (Cell Signaling, Danvers, MA, USA). Cells were sonicated using an ultrasonic homogenizer (10 s, 4 °C), followed by centrifugation of the homogenate (14000× *g*, 10 min, 4 °C). The bicinchoninic acid assay was performed to determine the protein concentration in the homogenates [23]. Samples containing 20 µg of protein were boiled for 5 min. After preparation, the samples were subjected to sodium dodecyl sulfate-polyacrylamide gel electrophoresis (SDS-PAGE) using 10% polyacrylamide gels [23]. After separation, the proteins were transferred onto 0.45-µm pore-sized nitrocellulose membranes (Amerscham, Buckinghamshire, UK) (35 V, 1.5 h, room temperature (RT)). Nonspecific binding was blocked via incubation (1 h, RT) in 0.05% (*v*/*v*) Tris-buffered saline–Tween-20 containing 1% bovine serum albumin (BSA; *v*/*v*) (Sigma, St. Louis, MO, USA). In all experiments, the membranes were cut horizontally according to the molecular weights of the investigated proteins. Membranes were incubated with specific primary antibodies against phosphorylated forms of Raf (1:1000), MEK (1:2000), p90 (1:1500), ERK (1:2000), and MSK (1:1000) (Cell Signaling), as well as a housekeeping protein/loading control GAPDH (1:10,000) in 1% BSA solution containing 0.1% Tween-20 (overnight, 4 °C). After washing (Tris-buffered saline containing 0.05% Tween-20, Sigma), the membranes were incubated with horseradish peroxidase-conjugated goat anti-rabbit secondary antibodies (Cell Signaling Technology) for 45 min at RT. A chemiluminescence analysis was performed using the LumiGLO 20× reagent (Cell Signaling), followed by exposure of the membranes to X-ray films. All films were scanned (Epson Perfection V19 Scanner, Epson, Suwa, Nagano, Japan) (8-bit, 400 dpi), and the density of the protein bands was quantified using Quantity One software (Bio-Rad) [24]. The 90-kDa form of MSK-1 was used for the calculation of the protein contents of the bands according to the protocol.

### 2.8. Apoptosis Assay and Flow Cytometry

Caspase 3/7 activity assays were performed using the FAM FLICA™ Caspase-3/7 Kit (Bio-Rad) combined with propidium iodide (PI). McCoy cells were divided into 4 experimental groups. Half of the McCoy cells were infected with 0.01 MOI *C. pneumoniae*, while the other half were left uninfected. Both the infected and uninfected groups were subdivided into those treated with 0.05-mg/mL Ax or left untreated. After 12 h of incubation, the apoptosis assay was performed according to the manufacturer’s instructions, and the cells were analyzed via flow cytometry. The assay discriminated between viable (caspase 3/7−/PI−), early apoptotic (caspase 3/7+/PI−), and late apoptotic (caspase 3/7+/PI+) cells. The fluorescence of the cell populations was analyzed immediately using a BD fluorescence-activated cell sorting Aria Fusion flow cytometer (BD biosciences).

### 2.9. In Vitro Effects of SP-A and SP-D on C. pneumoniae Proliferation and Attachment

McCoy cells were seeded (2.5 × 10^5^ cells) to form a monolayer in 24-well plates and incubated overnight. *C. pneumoniae* (0.01 MOI) was diluted in 1-mL medium in the presence of either SP-A (1 µg/mL) (Abcam, Cambridge, UK), SP-D (1 µg/mL) (Abcam), or neither. The mixtures were shaken at 37 °C for 1 h and then centrifuged at 800× *g* for 1 h; at 48 h pi, the cells were fixed using acetone and stained, as described in Section 2.1. To detect the effect of SP-A and SP-D proteins on the attachment of *C. pneumoniae* to J-774 cells (ATCC), the experiment described above was repeated with the following modification. After 1 h of centrifugation, the cells were stained immediately, as described earlier.

Fluorescence signals were analyzed using the Olympus IX83 Live Cell Imaging system and scanR High-Content Screening Station microscopy. The immunofluorescence of cells infected with nontreated *C. pneumoniae*, SP-A-pretreated *C. pneumoniae*, or SP-D-pretreated *C. pneumoniae* was analyzed quantitatively using ImageQuantTL 8.1; Image analysis software, GE Healthcare Bio-Sciences AB, Uppsala, Sweden, 2011 as follows: in three parallel cultures, 6–6 equally sized circular areas covering the cells were randomly selected on each image, and then, the background signals of the selected areas were eliminated via a threshold set-up, and the fluorescence intensity/pixel values of the randomly selected cells were quantitated.

### 2.10. Statistical Analysis

Welch’s *t*-test or one-way ANOVA RM with planned comparisons was performed using GraphPad Prism 8.0.1 software (GraphPad Software, San Diego, CA, USA). Data were expressed as the mean ± standard deviation. A value of *p* < 0.05 was considered significant.

## 3. Results

### 3.1. Ax Treatment at Elevated Concentration Suppressed C. pneumoniae Proliferation in Mouse Lungs

Previously, we found that Ax treatment at a therapeutic dose for humans does not affect the severity of *C. pneumoniae* infection in mice [17]. Our current aim was to determine whether a relatively higher concentration of Ax would suppress the chlamydia replication. Therefore, the mice were infected with *C. pneumoniae* (2 × 10^5^ IFU/mouse); half of the mice were treated daily from day 1 pi with a 4× higher concentration of Ax (5 mg/kg) than the dose used as a mucolytics in common respiratory infections. Seven days pi, the mice were euthanized, and the lungs were removed for detection of viable *C. pneumoniae*. A significant difference was found between the number of recoverable *C. pneumoniae* between the Ax-treated and untreated groups. The mean numbers of viable *C. pneumoniae* IFUs in the Ax-treated and untreated groups were 3.1 × 10^4^ and 7.3 × 10^4^ IFU/lung, respectively (Figure 1). The mice in the Ax-treated group exhibited less severe symptoms than those in the control group.

### 3.2. Ax Treatment Altered the Gene Expression and Protein Level of IFN-γ in the Lungs of C. pneumoniae-Infected Mice

We investigated whether gene expression was altered during Ax treatment, with respect to the cytokine profile in *C. pneumoniae* infection. We found that the relative expression of IL-12, IL-23, IL-17A, and IFN-γ was significantly higher in the *C. pneumoniae*-infected/Ax-treated group than in the *C. pneumoniae*-infected group without treatment (2.22-, 3.07-, 2.46-, and 2.27-fold, respectively) (Figure 2a).

We previously reported that Ax treatment elevates the expression of the known anti-chlamydial enzyme IDO-2 in the A-549 cell line; however, this elevation is not significant when a human equivalent mg/kg dose is administered to *C. pneumoniae*-infected mice [17]. Therefore, we determined whether an elevated dose of Ax increased IDO-1 and IDO-2 expression in mice. We observed no significant difference in expression levels between *C. pneumoniae*-infected mice and *C. pneumoniae*-infected/Ax-treated mice. However, in accordance with the findings of Virok et al., which showed that IDO-1 and IDO-2 are active in *C. pneumoniae*-infected mice [25], in contrast to earlier findings [26], we also found significant increases in both IDO-1 and IDO-2 expression levels during *C. pneumoniae* infection compared to those of the untreated/uninfected group (15.14- and 10.81-fold, respectively) (Figure 2a).

SPs play a crucial role in maintaining lung homeostasis; moreover, SP-D show an anti-*C. trachomatis* activity in genital mouse model [12,27,28,29]. Therefore, we determined the relative expression of SPs in the lungs of the mice. We found that all SPs were significantly upregulated in the untreated/infected mice compared to that of the uninfected/untreated group (SP-A, 4.55-fold; SP-B, 5.72-fold; SP-C, 5.28-fold; SP-D, 4.26-fold), suggesting that these anti-chlamydial mechanisms occur naturally in *C. pneumoniae*-infected mouse lungs. Notably, we observed a significant elevation in SP-A expression levels of mice in the treated/infected group compared to those of the untreated/infected group (2.89-fold) (Figure 2b).

After determining that the relative mRNA expression of *Ifng* was significantly higher in Ax-treated/infected mice than in untreated/infected mice, we investigated the protein expression levels of IFN-γ. We measured the cytokine concentrations in lung supernatants and found that IFN-γ levels were significantly higher in the *C. pneumoniae*-infected/Ax-treated group than in the untreated/infected group (*p* = 0.0041) (Figure 3). We also measured the level of the pro-inflammatory cytokine IL-6; however, we did not observe significant differences between the expression levels of *C. pneumoniae*-infected and *C. pneumoniae*-infected/Ax-treated groups (Figure 3).

### 3.3. SP Treatment Increased the Attachment of C. pneumoniae to Macrophages and Decreased Bacterial Proliferation

After showing that Ax treatment increased the relative expression of SPs in the lungs of *C. pneumoniae*-infected mice, we investigated the effect of anti-chlamydial SP proteins on *C. pneumoniae* replication and attachment to macrophages in vitro. We found that the pre-treatment of EBs with SP-A or SP-D significantly reduced the number of *C. pneumoniae* inclusions (0.18-fold) (*p* = 0.021) and (0.57-fold) (*p* = 0.034), respectively, in infected cells. This anti-chlamydial effect was more prominent in SP-A treatment than in SP-D treatment (Figure 4). These results correspond to the findings of Oberley et al. [12,29], who showed that SP-A and SP-D can aggregate *C. pneumoniae*, thereby inhibiting the infection of cells.

Oberley et al. demonstrated that SP-A-or SP-D-treated *C. pneumoniae* are recognized by THP-1 human monocytic cells with a higher efficiency than untreated bacteria [12]. Consequently, we investigated whether these findings were reproducible in J-774 murine macrophage cells. *C. pneumoniae* EBs were treated with SP-A or SP-D, and then, the samples were inoculated into J-774 cells. After 1 h of centrifugation, we stained the cells with *Chlamydia*-specific immunofluorescent antibodies, and it was found that the average pixel intensity of FITC was significantly elevated after treatment with both types of SPs compared to that seen in cells infected with untreated EBs. This indicated that a relatively larger number of *C. pneumoniae* cells were recognized by J-774 cells after SP treatment. This effect was more prominent in SP-A-treated *C. pneumoniae* than in SP-D-treated *C. pneumoniae* (SP-A 2.74-fold (*p* = 0.014)) and (SP-D 2.05-fold (*p* = 0.024) (Figure 5) (Appendix A).

### 3.4. Ax Treatment Did Not Induce Apoptosis via the Caspase-Dependent Pathway But Decreased ERK 1/2 Activation in C. pneumoniae-Infected Cells

It has been shown that *C. pneumoniae* can prevent the occurrence of host cell apoptosis; therefore, we investigated whether Ax treatment affected host cell death during *C. pneumoniae* infection. According to Galle et al., the assessment of apoptosis in *C. pneumoniae*-infected cells via annexin V staining is not accurate due to the *C. pneumoniae*-induced externalization of phosphatidylserine on the host cell membrane, which provides a binding site for annexin V and creates a false-positive apoptosis signal [30]. Therefore, we determined caspase 3/7 activity using flow cytometry. The analysis revealed that caspase 3/7 activity did not differ significantly between the treated/infected group and the untreated/infected group (Figure 6).

*C. pneumoniae* activates the ERK 1/2 pathway to acquire several essential molecules from host cells and avoid host cell apoptosis. Therefore, we investigated the effect of Ax treatment on the MAPK/ERK activity in *C. pneumoniae*-infected cells. Western blotting analysis showed that the ERK-1 (1.49-fold; *p* < 0.05) and ERK-2 (2.04-fold; *p* < 0.05) protein expression levels were significantly increased in untreated/infected cells compared to those in untreated/uninfected cells (Figure 7). We also observed that *C. pneumoniae* infection significantly increased the MSK-1 levels (2.25-fold) in untreated/infected cells compared to those in untreated/uninfected cells. MSK-1 is responsible for the activation of nuclear factor kappa-enhancer of B-cells and induction of early genes such as *c-fos*, *junB*, and *mkp-1* [31]. The Ax treatment of *C. pneumoniae*-infected cells significantly decreased the levels of c-RAF (0.42-fold), ERK 1 (0.16-fold), ERK 2 (0.15-fold), P90RSK1 (0.61-fold), P90RSK2 (0.67-fold), and MSK-1 (0.19-fold) (*p* < 0.05) compared to those of the untreated/infected group (Figure 7).

## 4. Discussion

Ax, a metabolite of bromhexine, is considered a relatively safe over-the-counter drug. It is primarily recommended as a secretory agent for the treatment of various respiratory diseases that are associated with extensive mucus production. Ax treatment increases surfactant synthesis and facilitates their secretion from type II pneumocytes [32,33]. Furthermore, Ax exhibits voltage-dependent sodium channel inhibitory properties, which can be beneficial in cases of diseases associated with sore throat by incorporating Ax in lozenges [8,34]. Ax can also be used in the treatment of Gaucher disease and is considered a supplemental medication for treating Parkinson’s disease, as it increases glucocerebrosidase activity [3,35]. Additionally, a recently published study suggested that when Ax is administered together with antibiotics, a relatively higher antibiotic concentration can be obtained in the lungs due to its secretion-supporting function [36]. Several studies have shown that Ax exhibits antimicrobial characteristics. Ax treatment reverses the resistance of *Candida albicans* to fluconazole [37]. Moreover, it inhibits the mucoid conversion of *Pseudomonas aeruginosa*, facilitates the bactericidal activity of ciprofloxacin against biofilms, and exhibits synergy with vancomycin for the elimination of catheter-related *Staphylococcus epidermidis* biofilms both in vitro and in vivo [38]. Furthermore, Ax impedes the rhinovirus infection in primary cultures of human tracheal epithelial cells [39]. Ax also shows antibiofilm properties; the biofilm formed by *Pseudomonas aeruginosa* treated with Ax for 7 days is thinner and more fragmented than that formed by untreated cells [40]. In our previous study, we confirmed the dose-dependent anti-chlamydial activity of Ax in *C. pneumoniae* infection in vitro; however, the commonly applied human dose does not affect *C. pneumoniae* proliferation in mice [17].

It is known that Ax has a well-balanced and favorable risk-benefit profile [2]; therefore, we investigated the effect of a four-fold increase in Ax dose on *C. pneumoniae* infection. In our in vivo study, we found that 5-mg/kg Ax significantly decreased the number of viable *C. pneumoniae* IFUs in the lungs of mice. IL-17A is associated with a neutrophil influx in the lungs; mice infected with *C. pneumoniae* and treated with anti-IL-17A antibodies have a relatively higher *C. pneumoniae* burden [41,42]. IL-23 has been described to be essential for inducing the *C. pneumoniae*-specific Th17 response [43]. In our study, Ax treatment in *C. pneumoniae*-infected mice significantly increased the relative expression levels of IFN-γ, IL-12, IL-17A, and IL-23 compared to those of the untreated/infected mice; additionally, the IFN-γ levels were higher in the lungs of the Ax-treated *C. pneumoniae*-infected mice than in infected/untreated mice. These results suggest that Ax treatment may enhance inflammation in the lungs and promote the anti-chlamydial response, thus resulting in a reduction in bacterial burden.

Furthermore, additional factors may contribute to a decreased number of recoverable *C. pneumoniae* inclusions. SP-A and SP-D are known to play an immunological role in maintaining lung homeostasis. It has been shown that these proteins are able to aggregate *C. pneumoniae* and *C. trachomatis* and can facilitate the uptake of bacteria by human macrophages [12,29]. Our RT-qPCR results suggest elevated SP-A levels that might also contribute to the improved elimination of bacteria in Ax-treated mice. Our in vitro results are in agreement with this phenomenon, as we found that the pretreatment of *C. pneumoniae* with SP-A or SP-D decreased the number of *C. pneumoniae* inclusion in the McCoy cell line, probably due the aggregating effect of SP-A and SP-D, thus resulting in a smaller proportion of *C. pneumoniae* infected cells. According to our results, SP-A and SP-D increased *C. pneumoniae* attachment to mouse macrophage cells, similar to the findings in human macrophage cells [12].

It is known that *C. pneumoniae* can inhibit cell apoptosis, and our results are in agreement with these findings, as we could not detect elevated caspase 3/7 activity at 12 h pi [10,13,44]. Additionally, *C. pneumoniae* stimulates the ERK 1/2 pathway activity to prevent caspase-independent apoptosis. Similarly, we found that *C. pneumoniae* infection significantly enhanced the ERK 1/2 pathway activity compared to that in uninfected/untreated cells [10,11]. Notably, we observed a novel possible mechanism via which *C. pneumoniae* prevents host cell apoptosis: *C. pneumoniae* significantly elevates the levels of phosphorylated MSK-1, which can also promote BCL2-associated agonist of cell death (BAD) phosphorylation and lead to the repression of BAD-induced apoptosis [45,46]. When the cells were treated with Ax in *C. pneumoniae*-treated cells, MAPK/ERK activity was significantly reduced. This result suggests that the anti-chlamydial activity of Ax may be partially attributed to the reduced MAPK/ERK 1/2 activity. The MAPK/ERK 1/2 pathway also plays a crucial role in the nourishment of proliferating bacteria, and inhibition of this process is fatal to the pathogen; it has been shown that MAPK inhibitors can inhibit *C. pneumoniae* infection [11,47]. The level of the P90RSK1/2 protein was also found to be significantly decreased; this protein is crucial for the phosphorylation of several antiapoptotic proteins such as death-associated protein kinase 1 and BAD [48]. Additionally, the levels of phosphorylated MSK-1 decrease significantly, which are known to result in apoptosis. These findings suggest that Ax treatment may induce apoptosis in *C. pneumoniae*-infected cells, thereby inhibiting bacterial proliferation.

The results of a phase 2 clinical trial showed that Ax is well-tolerated beyond its standard administration at 1.25 mg/kg [49]. Given the coronavirus disease 2019 (COVID-19) pandemic, several studies have shown that Ax can be used as a supplemental medication for treating COVID-19 [50,51,52]. Notably, MAPK/ERK activation plays a pivotal role in additional bacterial infections, including *Coxiella burnetii* infections, which are similar to *Chlamydia* infections. Therefore, we think it may be worth to investigate the role of Ax in *C. burnetii* infections.

## 5. Conclusions

Ax has been used as a medication since 1980 for treating respiratory infections; however, it was shown that Ax, at a normal dose used for humans, does not inhibit *C. pneumoniae* proliferation in mice. We found that *C. pneumoniae*-infected mice treated with four times higher doses of Ax than normal contained significantly fewer viable *C. pneumoniae* IFUs than those in untreated mice. The mRNA expression levels of IL-12, IL-23, IL-17F, IFN-γ, and SP-A were significantly increased in infected mice treated with Ax compared to those in control mice. Moreover, we found that Ax treatment decreased the activity of the MAPK/ERK pathway, which may have induced apoptosis in infected cells. Based on our results, a higher dose of Ax exhibits an anti-chlamydial effect in mice, probable due to its modulating effects on the immunologic and metabolic pathways.

## Figures and Tables

**Figure 1 microorganisms-09-00880-f001:**
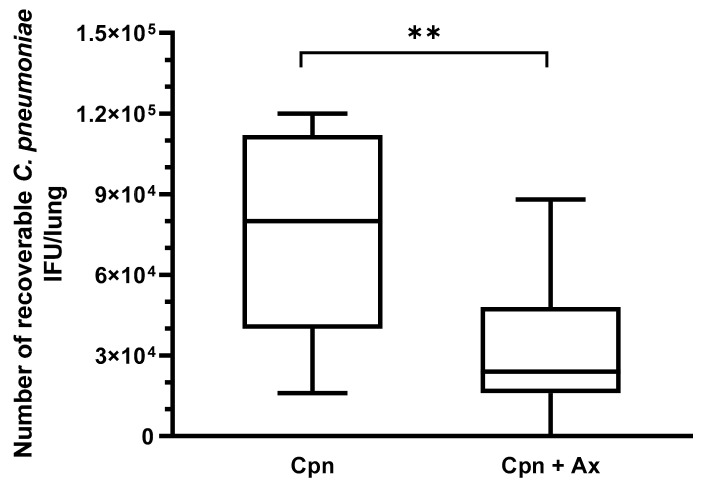
Recoverable *C. pneumoniae* IFU in infected mice (*n* = 20) with and without Ax treatment. Ax-treated mice daily received 5-mg/kg Ax for 6 days. The supernatant of the lung homogenates was inoculated into a McCoy cell monolayer and cultured for 2 days, followed by fixation and staining, and the number of inclusions was determined via indirect immunofluorescence assay. (** *p* = 0.0049). Box = 25th percentile, median, and 75th percentiles; bars = min and max values. IFU, inclusion-forming unit; Cpn, *C. pneumoniae*-infected mice; Cpn + Ax, *C. pneumoniae*-infected mice treated with Ambroxol.

**Figure 2 microorganisms-09-00880-f002:**
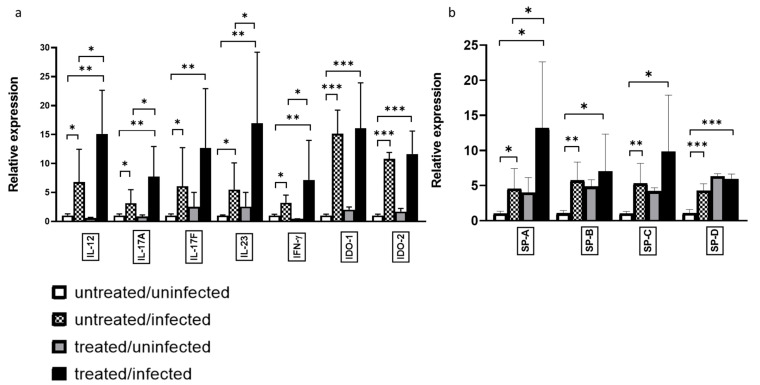
Relative expression of genes influences by *C. pneumoniae* infection and Ax treatment. Mice were either infected with *C. pneumoniae* (or remained uninfected as a control), followed by treatment with Ax (5mg/kg) (or remained untreated, for the control). RNA was extracted from the lungs, and gene expression was analyzed via RT-qPCR for (**a**) IL-12, IL-23, IL-17A, IL-17F, IFN-γ, IDO-1, and IDO-2 and (**b**) SP-A, SP-B, SP-C, and SP-D. Bars denote the mean and standard deviation of the expression levels for triplicate measurements (* *p* < 0.05, ** *p* < 0.01, and *** *p* < 0.001).

**Figure 3 microorganisms-09-00880-f003:**
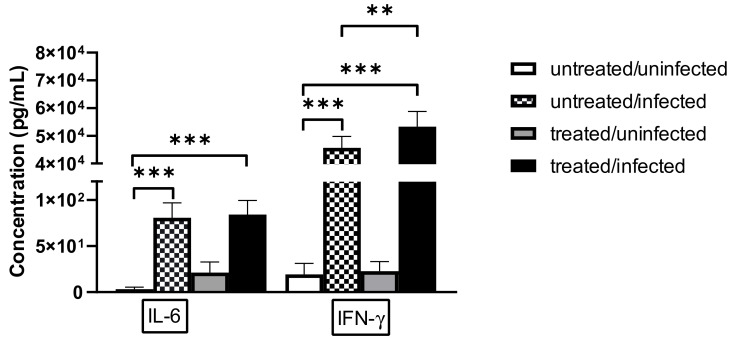
Concentrations of IL-6 and IFN-γ in lung supernatants. Mice were infected intranasally with *C. pneumoniae* and treated with 5-mg/kg Ax for 6 days or left untreated and uninfected. On the seventh day post-infection, the mice were euthanized and the concentrations of IL-6 and IFN-γ in the supernatants of the lungs were measured via ELISA. Each bar denotes the mean ± standard deviation for 10 mice (** *p* < 0.01, and *** *p* < 0.001).

**Figure 4 microorganisms-09-00880-f004:**
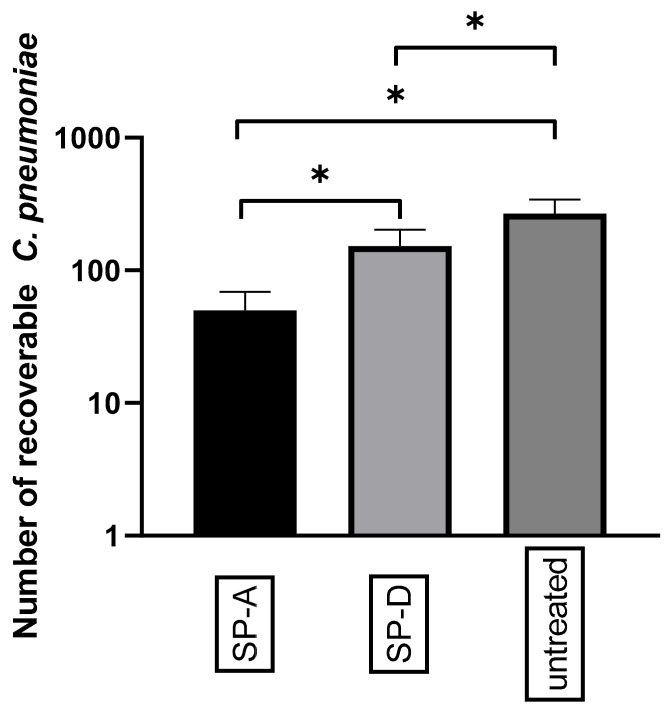
Anti-chlamydial effect of SP-A and SP-D. *C. pneumoniae* EBs were treated with either SP-A, SP-D, or left untreated. After 1 h of incubation, the samples were inoculated into McCoy cells and fixed 48 h post-infection. The number of *C. pneumoniae* inclusions was determined via indirect immunofluorescence staining. Each bar denotes the mean number of inclusions in three parallel cultures ± standard deviation (* *p* < 0.05).

**Figure 5 microorganisms-09-00880-f005:**
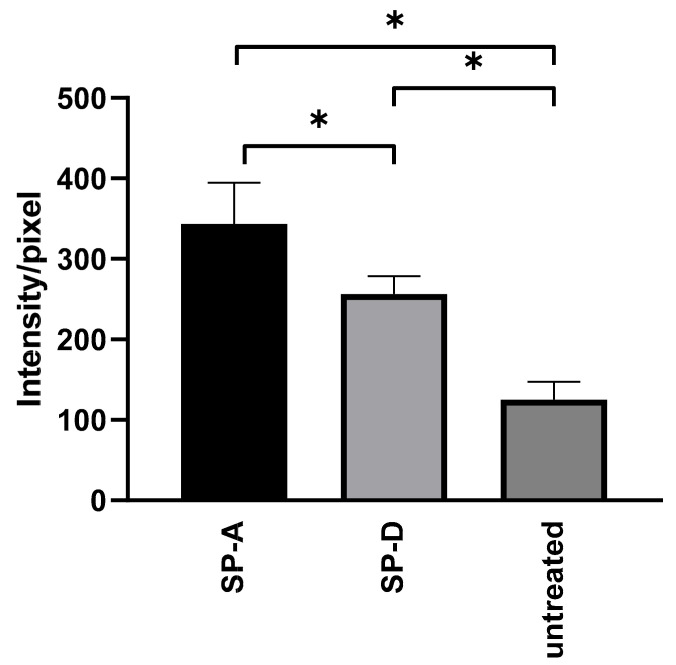
Effect of SP-A and SP-D treatment on *C. pneumoniae* attachment to J-774 cells. *C. pneumoniae* EBs were incubated with either SP-A or SP-D, or they were left untreated. After 1 h of incubation, the samples were inoculated into J-774 cells, and then, 1 h post-infection, the cells were fixed and stained via immunofluorescence. The average FITC intensity/pixel was measured. The results are expressed as the mean ± standard deviation of the data from three independent experiments. Microscopic pictures are shown in Appendix A, (* *p* < 0.05).

**Figure 6 microorganisms-09-00880-f006:**
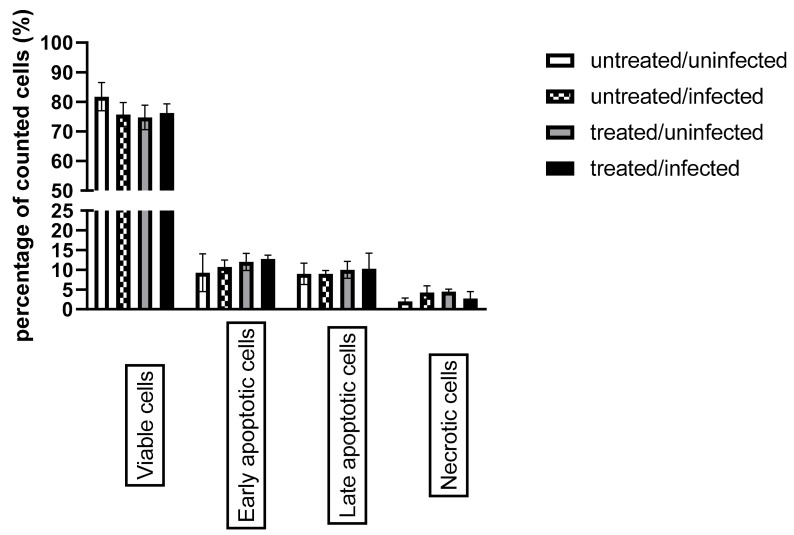
Evaluation of host cell apoptosis based on caspase 3/7 activity combined with propidium iodide staining. McCoy cells were infected with 0.01 MOI of *C. pneumoniae* and/or treated with 0.05-g/mL Ax or left untreated/uninfected. Flow cytometric analysis was performed 12 h post-infection to determine the status of apoptosis. The results are expressed as the mean ± standard deviation of the data from three independent experiments. (*p* > 0.05).

**Figure 7 microorganisms-09-00880-f007:**
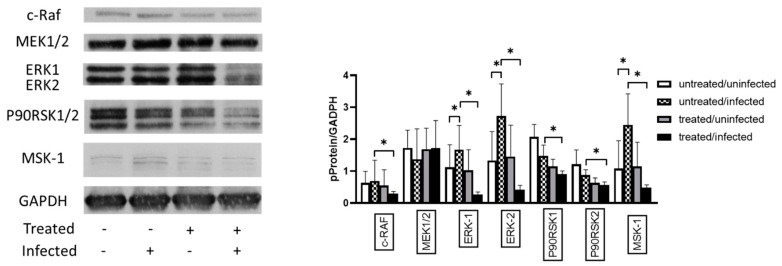
Analysis of the expression of proteins related to the MAPK/ERK pathway. McCoy cells were infected with 0.01 MOI *C. pneumoniae*, treated with 0.05-mg/mL Ax, infected with 0.01 MOI *C. pneumoniae* and treated with 0.05-mg/mL Ax, or left untreated. The cells were scraped at 12 h post-infection. Subsequently, proteins in the cell lysates were separated via SDS-PAGE and then blotted onto nitrocellulose membranes. Membranes were incubated with specific primary antibodies against phosphorylated forms of c-RAF, MEK1/2, P90RSK1/2, ERK1/2, MSK-1, and a housekeeping protein/loading control GAPDH. Horseradish–peroxidase-conjugated goat anti-rabbit IgG was used as secondary antibody. Chemiluminescence analysis was performed using the LumiGLO reagent, followed by the exposure of membranes to X-ray films. All films were scanned, and density of the protein bands was quantified using the Quantity One software. The ratio of phosphorylated protein/GAPDH was determined. The results are expressed as the mean ± standard deviation of the data from three independent experiments. The shown Western blot is representative of parallel Western blots (* *p* < 0.05). MAPK, mitogen-activated protein kinase; ERK, extracellular signal-regulated kinase; c-RAF, RAF proto-oncogene serine/threonine-protein kinase; MEK1, dual specificity mitogen-activated protein kinase; MSK-1, mitogen- and stress-activated protein kinase 1; GAPDH, glyceraldehyde 3-phosphate dehydrogenase.

## Data Availability

Not applicable.

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
