# Peer review of "Ambroxol Treatment Suppresses the Proliferation of Chlamydia pneumoniae in Murine Lungs"

_microorganisms, 2021, doi:10.3390/microorganisms9040880_

Round 1

Reviewer 1 Report

The manuscript on “Ambroxol treatment suppresses the proliferation of Chlamydia pneumoniae in murine lungs” intends to investigate the mechanism by which the drug Ambroxol promote anti-chlamydial activity. The manuscript is as such written with very less introduction and the purpose of the study is not well defined. The introduction is very vague and substantial information on the objective of the research is not given. Ax has an anti-chlamydial activity in human but in mice it did not show any effect (at the particular concentration given). It is not explained why the authors are interesting and are addressing the effect of the drug in the mice.  

Major comments:

  1. In Figure 1 the authors clearly show that Ax has a significant effect in anti-chlamydial activity in mouse lung. This data is contradictory to what the authors claim in the introduction.
  2. Since the concentration of Ax is very critical for its activity, the amount of Ax used in Figure 2a-b is important to be mentioned.
  3. It is not clear how Ax addition leads to increase in cytokines or SP
  4. How is SP-A more efficient than Sp-D?
  5. In Figure 6 the authors show that Ax has no significant effect in regulating the apoptotic pathway, so how does this explain the anti-chlamydial activity?
  6. What is the relevance of Figure 7?, Chlamydia is known to activate the MAPK pathway. The western blots shown in Figure 7 do not show ERK phosphorylation, rather it looks reduced in these blots. Moreover, there is no infection marker (eg. cHSP60, or any bacterial protein) to show that there was significant infection in the infected lanes.

The authors had taken the effort to investigate how the drug Ax work but, there is no significant data or finding showing how Ax regulates or blocks Chlamydial infection.

Author Response

Dear Reviewer,

Reviewer 2 Report

This is a well-written and well-designed research article looking at the therapeutic potential of Ambroxol on Chlamydia pneumoniae infection.  Authors investigate the effects on recoverable CFUs, gene expression of various cytokines.  SP-A and SP-D treatment is also investigated. Only minor revisions are requested by this reviewer.

In Figures 1-6, please define error bars. Are they standard deviation or standard error of the mean? Add this detail to each figure legend

Line 47: Define SP

Line 157: Can the authors confirm the MOI being 0.01?  This is incredibly low for Chlamydia infections and if this is correct, why was such a low MOI chosen for these experiments?

Lines 208 and 209: C. pneumoniae should be in italics

Author Response

Dear Reviewer,

Please see the attachment,

Reviewer 3 Report

The manuscript ‘Ambroxol treatment suppresses the proliferation of Chlamydia pneumoniae in murine lungs’ examines the impact of Ambroxol treatment on Chlamydia pneumoniae colonisation of murine lungs and the modulation of host immune responses. This manuscript builds on the authors previous publication, which demonstrates that standard therapeutic dose of Ambroxol does not affect C. pneumoniae colonisation. The authors have therefore examined the impact of a dose 4x that of the standard, showing that this concentration of Ambroxol does reduce the colonisation of C. pneumoniae in the murine lungs and alters the gene expression and protein levels of key innate immune cytokines and signalling pathway proteins. Overall the manuscript is well written and experiments are appropriately controlled throughout, with descriptive methods. However, I feel that the manuscript would benefit from addressing the following minor comments.

  1. Lines 185-187 – Please consider rephrasing this statement to make it more conscience.
  2. Line 224 – Consider rephrasing the statement “..beneficial effect on chlamydia replication” the current wording implies that this would be beneficial to the bacteria and not in terms of controlling the infection.
  3. Lines 225 and 226 – Please include the IFU and mg/kg doses for pneumoniae and Ambroxol, respectively.
  4. Figure 1 –
    1. The data should be presented on a log scale.
    2. Line 238 - Please include the actual P value calculated.
  5. Figure 2 –
    1. Check the figure labelling and the corresponding text. Currently figure 2b in the text is actually figure 2c in the image.
    2. Could figures 2a and 2c (image) be combined to make a single graph? Is there a reason why these results have been separated? As the authors discuss the results from both figures together, so I’m unclear why they have been separated into different panels.
    3. Please ensure the text size and font are consistent across all panels, currently 2b is clearly larger than 2a or 2c.
    4. Are all the statistically significant results between <0.05 and 0.01? I recommend the authors consider using a scale to show enhanced significance between different samples (ie * <0.05, **<0.01, ***<0.001, etc)
  6. Line 278 – Please provide the actual P value
  7. While I appreciate this maybe personal preference, perhaps the authors could consider combining figures 2a/2c and figure 3 into a single multi-panel figure. While combining figures 2b and figure 4 into a single figure, thereby grouping the analysis of components together rather than figures based on the assay being implemented.
  8. Figure 4 –
    1. The data should be presented on a log scale.
    2. Please provide the actual P value.
  9. Figure 5 –
    1. Please provide the actual P value.
    2. Please include representative microscopy images and/or all the images used for quantification in supplementary data.
  10. Figure 6 –
    1. Please clarify what % refers to, ie % cell survival/ negative staining?
  11. Figure 7 –
    1. Please provide a clear explanation in the corresponding text (ie Lines 339-351) as to how the protein concentrations were quantified. As in some instances the representative western blot images do not correlate with the graphical representation or the conclusions which have been drawn. For example, the ERK 1 band is not increased in concentration in the untreated/infected sample compared to untreated/uninfected control. However, this is in contrast to the statement on lines 343-344.
    2. The MSK-1 blots show multiple bands, please confirm which is the correct band (ie used for concentration calculations) and secondly, is the antibody specific to only the phosphorylated versions of this protein? Please clarify this in the text.
    3. In the interests of data transparency, I strongly recommend the authors include the raw (densitometry) data used to generate the graphical representation in Figure 7 as the MEK1/2 data presented in the graph doesn’t appear to correlate with that of the representative western blot image.
  12. Line 403 and 409– Please rephrase to remove/replace the word “elimination” technically your data doesn’t show “elimination” it shows a reduction in bacterial burdens.
  13. Line 407 – Is anything known regarding the impact of SP-A and SP-D on phagocytosis by neutrophils? Given your data shows enhancement of IL-17 and IL-23, known drivers of neutrophil recruitment.
  14. Line 411 – “SP-A or SP-D decreased the proliferation ....” was this reduced proliferation of the pathogen or reduced internalisation of the EB? Your data doesn’t differentiate between these two possibilities, please rephrase the statement to clarify.
  15. Line 419 – The statement “ C. pneumoniae significantly elevates the levels of phosphorylated MSK-1” – see my earlier comment regarding your MSK-1 antibody.
  16. Line 433/434 – “Ax is well-tolerated beyond its standard administration” could the authors expand on this statement. What doses have been tested in humans, how applicable is the dose described here for the treatment of human infections, is there any toxicity data available concerning high doses?

Author Response

Dear Reviewer,

Round 2

Reviewer 1 Report

The authors have explained the reviewer comments satisfactorily. But it is still not clear with figure 7. Chlamydia is known to activae MAPK pathway, evident in phopshorylation of ERK and AKT. In figure 7 the authors have only shown the levels of total ERK and AKT. As the authors claim that Ax is it is important MAPK activity, it is critical to show the phosphorylation status and also qauntify the blots via densitometry. 

Author Response

Dear Reviewer,

I would like to thank you for your comments, and I am addressing the questions as follows.

The authors have explained the reviewer comments satisfactorily. But it is still not clear with figure 7. Chlamydia is known to activae MAPK pathway, evident in phopshorylation of ERK and AKT. In figure 7 the authors have only shown the levels of total ERK and AKT. As the authors claim that Ax is it is important MAPK activity, it is critical to show the phosphorylation status and also qauntify the blots via densitometry. 

                In the Materials and Methods section, “2.7. Western blotting analysis of ERK” subsection in Line 173 we describe that the primary antibodies used during the Western blot analysis reacted only with the phosphorylated form of the proteins. “Membranes were incubated with specific primary antibodies against phosphorylated forms of Raf (1:1000), MEK (1:2000), p90 (1:1500), ERK (1:2000), and MSK (1:1000) (Cell Signaling)”. This information is given in the legend of Figure 7 too.

Furthermore, membranes were exposed to X-ray films and the quantitative evaluation of the protein bands on scanned films was done via densitometry, as described in Lines 180-182: „All films were scanned (Epson Perfection V19 Scanner, Epson, Suwa, Nagano, Japan) (8-bit, 400 dpi), and the density of protein bands was quantified using Quantity One software (Bio-Rad) [24].” The legend to Figure 7 also contain this information. The column chart of Figure 7 was produced using the obtained data.

                In this study we aimed to examine the effect of Ax-treatment on the level of proteins related to ERK pathway during C. pneumoniae infection. We did not investigate the level of proteins related to AKT pathway.